# Chaetomadrasins A and B, Two New Cytotoxic Cytochalasans from Desert Soil-Derived Fungus *Chaetomium madrasense* 375

**DOI:** 10.3390/molecules24183240

**Published:** 2019-09-05

**Authors:** Qing-Feng Guo, Zhen-Hua Yin, Juan-Juan Zhang, Wen-Yi Kang, Xue-Wei Wang, Gang Ding, Lin Chen

**Affiliations:** 1Henan Joint International Research Laboratory of Drug Discovery of Small Molecules, Zhengzhou Key Laboratory of Synthetic Biology of Natural Products, Huanghe Science and Technology College, Zhengzhou 450063, Henan, China (Q.-F.G.) (Z.-H.Y.) (J.-J.Z.) (W.-Y.K.); 2Institute of Microbiology, Chinese Academy of Science, Beijing 100101, China; 3Institute of Medicinal Plant Development, Chinese Academy of Medical Science and Union Medical College, Beijing 100193, China

**Keywords:** *Chaetomium madrasense*, cytochalasans, cytotoxicity, fungal alkaloids

## Abstract

Two new cytochalasans, Chaetomadrasins A (**1**) and B (**2**), along with six known analogues (**3**–**8**), were isolated from the solid-state fermented culture of desert soil-derived *Chaetomium madrasense* 375. Their structures were clarified by comprehensive spectroscopic analyses, and the absolute configurations of Compounds **1** and **2** were confirmed by electronic circular dichroism (ECD) and calculated ECD. For the first time, Chaetomadrasins A (**1**), which belongs to the chaetoglobosin family, is characterized by the presence of all oxygen atoms in the form of Carbonyl. Chaetomadrasin B (**2**) represents the first example of chaetoglobosin type cytochalasan characterized by a hydroxy unit and carbonyl group fused to the indole ring. Compounds **1** and **2** displayed moderate cytotoxicity against HepG2 human hepatocellular carcinoma cells.

## 1. Introduction

Cytochalasans are a well-known class of alkaloids characterized by a perhydroisoindolone moiety, to which a typical macrocyclic ring is fused [1]. These fungal alkaloids are characterized by a polyketide backbone and an amino acid (such as leucine, tryptophan or phenylalanine) with a broad spectrum of bioactivity, including cytotoxic [2,3,4], antibacterial [5], phytotoxic [6], antiviral [7], and immunomodulatory activities [8,9]. Cytochalasans caught chemists’ and pharmacologists’ attention because of their complex polycyclic fused skeletons and interesting biological activities. Numerous bioactive cytochalasans with novel skeletons have been reported in recent years [10,11,12,13], and, to date, more than 300 cytochalasans or analogues have been reported from diverse fungal genera, including *Chaetomium*, *Aspergillus*, *Trichoderma*, and *Periconia* [12,13,14,15].

In our continued discovery of bioactive natural products from the members of special fungi isolated from the desert and grasslands inhabiting the Northwest of China [16,17,18,19], two new cytochalasan derivatives, Chaetomadrasins A (**1**) and B (**2**), together with six related known compounds (**3**–**8**) (Figure 1), were isolated and identified from the ethyl acetate extract of a solid-state fermented culture of *Chaetomium madrasense* 375, which was collected from desert soil in Hotan city, Sinkiang province, People’s Republic of China. However, as far as we know, this is the first report on secondary metabolites from the fungi. Chaetomadrasins A (**1**) and B (**2**) were evaluated in vitro for their cytotoxicities against the HepG-2 cell line, with *cis*-platin as a positive control. Herein, we present the isolation, structural elucidation, and bioactivity of these compounds.

## 2. Results and Discussion

Chaetomadrasin A (**1**) was isolated as a white amorphous powder. The HR-ESI-MS data suggested a molecular formula of C_32_H_36_N_2_O_5_ based on the [M + Na]^+^ ion signal at 551.2518. The IR spectrum showed absorption bands at 3370 and 1714 cm^−1^, thereby implying the presence of amino and carbonyl groups. The aromatic protons signals at *δ*_H_ 7.51 (d, 7.0, H-4′), 7.15 (t, 7.4, H-5′), 7.22 (t, 7.5, H-6′), and 7.36 (d, 8.0, H-7′), along with an olefinic proton at *δ*_H_ 7.09 (s, H-2′) and a broad NH singlet at *δ*_H_ 8.71 (H-1′), could be assigned to a 3-substituted indolyl group. The ^1^H, ^13^C (Table 1), and HSQC (Appendix A) nuclear magnetic resonance (NMR) spectral data for **1** revealed the presence of four methyl groups, four methylene units, six methine units, one quaternary carbon, 12 olefinic and aromatic carbons, and five carbonyl carbons, which were quite similar to those of chaetoglobosin Y [20]. The main differences between the two compounds are at positions C-20, with the hydroxy substituent (C-20) in chaetoglobosin Y being replaced by the carbonyl group on C-20 in Compound **1**. This conclusion is further supported by the chemical shifts of C-20 (*δ*_C_ 204.8) and the ^1^H detected heteronuclear multiple bond correlation (HMBC) (Appendix A) cross-peaks from H-21 to C-20. Further evidence for the structure of **1** was provided by its HMBC and ^1^H-^1^H correlation spectroscopy (COSY) spectra (Figure 2). In this way, the planar structure of **1** was characterized. The relative configuration of **1** was established by nuclear overhauser enhancement spectroscopy (NOESY) experiment. In the perhydro-isoindolone ring, the NOESY (Appendix A, Figure 3) correlations of 11-Me/H-3, H-4/H-8, H-4/H-10a, H-5/H-8, and H-6/H-8 implied that H-3, 11-Me, and 12-Me are α-oriented, while H-4, H-5, H-6, and H-8 are *β*-oriented. Furthermore, NOESY correlations from H-14 to H-8 and H-16 suggested that H-14, H-8 and H-16 were cofacial and *β*-oriented. Furthermore, the large coupling constant of H-8/H-13 (*J*_8,3_ = 15.1 Hz) suggested that H-13 should be *α*-oriented.

To the best of our knowledge, only chaetoglobosin Y and chaetoglobosin Z [21] possess the same perhydro-isoindolone moiety as that of **1** among all known chaetoglobosins. However, their relative configurations at C-6 were different. Therefore, the absolute configuration of **1** was expected to be identical with that of chaetoglobosin Y or chaetoglobosin Z. As a result, absolute conformational analyses of (3*S*, 4*R*, 5*S*, 6*S*, 8*R*, 9*R*, 16*S*)-**1** and (3*S*, 4*R*, 5*S*, 6*R*, 8*R*, 9*R*, 16*S*)-**1** were performed using time-dependent density functional theory (TDDFT)-ECD calculations. The results (Figure 4) indicated that the calculated ECD curve of (3*S*, 4*R*, 5*S*, 6*S*, 8*R*, 9*R*, 16*S*)-**1** matched well with the experimental one. In this way, the structure of **1** was elucidated (as shown in Figure 1) and named Chaetomadrasin A.

Chaetomadrasin B (**2**) was also obtained as a white amorphous powder. The molecular formula of **2** was suggested to be C_32_H_36_N_2_O_7_ on the basis of HR-ESI-MS ([M + H]^+^ at *m/z* 561.2590). Detailed analysis of the ^1^H and ^13^C NMR spectra (Table 1) of **2** indicated that this compound was also a chaetoglobosin derivative. The unexpected hydroxy and carbonyl groups fused to the indole ring in **2** were evidenced by the chemical shifts of C-2′ (*δ*_C_ 178.8) and C-3′ (*δ*_C_ 74.6), as well as the HMBC correlations from H-10 to C-2′, C-3′, H-1′ to C-3′, H-3 to C-3′and H-4′ to C-3′ (Appendix A). The complete structure of **2** was determined by correlative analysis of the 2D NMR spectra (Figure 3, Appendix A) and comparing the NMR data with chaetoglobosin V_b_ (**6**) [22] and cytoglobosin A(**7**) [23]. The observed ROESY correlations (Figure 3, Appendix A) of H-3/Me-11; H-7/H-13; H-4/H-10; H-8/H-14; H-16/H-21; H-13/H-17 closely resembled those of chaetoglobosin V_b_ (**6**), indicating the same relative configuration. The identical biosynthetic pathways, relative configurations, and similar ECD spectra (Figure 2) between **2** and chaetoglobosin V_b_ (**6**) suggests that both compounds share absolute configurations, except for their indole ring moieties. Subsequently, the ECD experiments and ECD calculations (Figure 2) of **2** were applied to confirm the absolute configuration of C-3′. The absolute configuration of **2** was established as (3′*R*, 3*S*, 4*R*, 7*S*, 8*R*, 9*R*, 16*S*, 17*R*, 21*R*)-**2** by comparing its experimental and theoretical ECD spectra. Therefore, the structure of **2** was constructed as Chaetomadrasin B.

Six known cytochalasan alkaloids were identified as chaetoglobosin G (**3**) [24,25], isochaetoglobosin D (**4**) [23], armochaetoglobin U (**5**) [26], chaetoglobosin V_b_ (**6**) [22], cytoglobosin A (**7**) [23], and cytoglobosin A_b_ (**8**) [27] by comparing their NMR and HR-ESI-MS data with those in the scientific literature.

To make sure that **2** is not the reduction product of **1** during the experimental process, the crude extract was compared to that of pure compounds **1** and **2** via an HPLC chromatogram (see Appendix A). The result indicated that **1** and **2** are the products of *C. madrasense* 375. The cytotoxicity of **1** and **2** against the HepG-2 cell line were evaluated by using the Cell Counting Kit-8 (CCK-8) method. Compounds **1** and **2** showed moderate cytotoxicity against HepG2 human hepatocellular carcinoma cells with an IC_50_ of 8.7 and 19.4 μM, respectively. The IC_50_ value of the positive control (*cis*-platin) was 3.14 μM.

## 3. Experimental Section

### 3.1. General Experimental Procedures

Optical rotations were tested on a Perkin-Elmer 241 polarimeter (Waltham, MA, USA), UV data were recorded on a Hitachi U-4100 UV-Vis spectrophotometer (Hitachi co., Ltd, Tokyo, Japan), and the IR spectra (KBr) were recorded on a Thermo Scientific NICOLET-iS5 FT-IR spectrometer (Thermo, San Jose, CA, USA). ECD spectra were acquired by a JASCO J-815 spectrometer (JASCO Corporation, Tokyo, Japan). NMR spectroscopic data were acquired in CDCl_3_ or DMSO-*d*_6_ by using a Bruker AVANCE III 400 NMR spectrometer (Bruker, Billerica, MA, USA) with tetramethylsilane (TMS) or solvent signals (CDCl_3_; *δ*_H_ 7.26/*δ*_C_ 77.0; DMSO-*d*_6_; *δ*_H_ 2.50/*δ*_C_ 40.0) as the internal reference. HR-ESI-MS were recorded on an Agilent 6250 TOF LC/MS (Agilent Technologies, Santa Clara, CA, USA). The semi-preparative HPLC was performed on a Calmflow^plus^ instrument packed with a YMC Pack ODS-A column (10 mm × 250 mm 5 μm, Japan) and a 50D UV-vis Detector (Lumiere Tech Ltd., Berlin, Germany). Column chromatography was carried on silica gel (200–300 mesh; Qingdao Marine Chemical Factory, Qingdao, China), reverse phase (ODS) silica gel (YMC, Tokyo, Japan), or Sephadex LH-20 (GE Healthcare BioSciences AB, Uppsala, Sweden). The chemicals used in the study were of analytical grade.

### 3.2. Fungal Material

The title fungus, strain number 375 (CCTCC M2019517 CLC375), was collected from a soil sample obtained in Hotan city, Sinkiang province, People’s Republic of China. The strain was identified by Dr XueWei Wang as *Chaetomium madrasense* based on its morphological character, as well as the 18s rDNA sequence. The strain’s sequences were deposited in the GenBank as KP269060.1. The strain was cultured on potato dextrose agar at 25 °C for 7 days as the seed culture. Agar plugs were cut into small pieces (approximately 1 cm × 1 cm) and inoculated into 300 Erlenmeyer flasks (500 mL), previously sterilized by autoclaving, each containing 60 g of rice and 100 mL of distilled water. All flasks were incubated at 25 °C for 28 days.

### 3.3. Extraction and Isolation

After incubation, the fermented material was extracted by ethyl acetate (3 times) at room temperature, and the solvent was evaporated until it was dry under reduced pressure to produce a brown crude extract (200.0 g). The extract was fractionated by silica gel column chromatography eluted with CH_2_Cl_2_/CH_3_OH (100:0–1:1 *v*/*v*) to give six fractions (Fr.1–Fr.6). Fr. 4 (40.2 g) was subjected to silica gel column chromatography, eluted with petroleum ether/acetone (50:1–10:1 *v*/*v*) to give eight fractions (Fr. 4.1–Fr. 4.8). Fr. 4.5 (2.8 g) was further separated by Sephadex LH-20 (CH_2_Cl_2_/MeOH **v*/*v** 1:1) to yield five subfractions (Fr.4.5.1–Fr.4.5.5). Fr.4.5.3 (130 mg) was then purified by semipreparative HPLC (50% MeCN in H_2_O) to obtain chaetoglobosin V_b_ (**6**; 39.5 mg, *t*_R_ 15.0 min). Fr.4.6 (6.5 g) was subjected to silica gel column chromatography and eluted with CH_2_Cl_2_/MeOH (50:1–10:1 **v*/*v**) to get seven subfractions (Fr.4.6.1–Fr.4.6.7). Fr.4.6.2 (300 mg) was further separated by Sephadex LH-20 (MeOH) and semipreparative HPLC (50% MeCN in H_2_O) to obtain chaetoglobosin G (**3**; 16.5 mg, *t*_R_ 23.6 min). Fr.4.6.3 (1.12 g) was further separated by Sephadex LH-20 (100% MeOH) and semipreparative HPLC (50% MeCN in H_2_O) to obtain Chaetomadrasin A (**1**; 8.6 mg, *t*_R_ 33.9 min) and isochaetoglobosin D (**4**; 4.5 mg, *t*_R_ 22.9 min). Fr.4.6.4 (923 mg) was further purified by repeated silica gel column chromatography, Sephadex LH-20 chromatography (MeOH), and semipreparative HPLC (45% MeCN in H_2_O) to obtain cytoglobosin A (**7**; 2.0 mg, *t*_R_ 27.8 min). Fr.5 (1.7 g) was purified by silica gel column chromatography (CH_2_Cl_2_/CH_3_OH, **v*/*v** 10:1) and Sephadex LH-20 (100% MeOH), followed by semipreparative (35% MeCN in H_2_O) to give cytoglobosin A_b_ (**8**; 1.6 mg, *t*_R_ 37.5 min). Fr.6 (4.6 g) was subjected to ODS column eluted with CH_3_OH/H_2_O (20%–100%, *v*/*v*) to give 5 fractions (Fr. 6.1–Fr. 6.5). Fr.6.1 (500 mg) was further separated by Sephadex LH-20 (MeOH) and semipreparative HPLC (45% MeCN in H_2_O) to obtain armochaetoglobin U (**5**; 3.5 mg, *t_R_* 27.0 min). Fr.6.2 (802 mg) was further separated by Sephadex LH-20 (MeOH) and semipreparative HPLC (30% MeCN in H_2_O) to obtain chaetomadrasin B (**2**; 5.2 mg, *t*_R_ 30.7 min).

Chaetomadrasin A (**1**): white amorphous powder; [α]^25^_D_ = −8.5 (c 0.2, MeOH); UV (MeOH) λ_max_ (log*ε*) 253(0.83), 221 (2.82), 216 (2.81) nm; IR (KBr) *υ*_max_ 3370, 2926, 1714, 1458, 746 cm^−1^; ECD(MeOH) λ_max_ (Δ*ε*/M^−1^cm^−1^) = 298 (−1.2), 267 (+0.8), 243 (−0.6), 207 (+4.1) nm; for ^1^H NMR, ^13^C NMR, see Table 1; HR-ESI-MS *m/z* 551.2518 [M + Na]^+^ (calcd for C_32_H_36_N_2_O_5_, 551.2522 [M + Na]^+^).

Chaetomadrasin B (**2**): white amorphous powder; [α]^25^_D_ = −12.5 (c 0.2, MeOH); UV (MeOH) λ_max_ (log*ε*) 254 (2.68), 229 (3.22), 213 (3.08) nm; IR (KBr) *υ*_max_ 3392, 3261, 2956, 1701, 1647,1470 cm^−1^; ECD(MeOH) λ_max_ (Δ*ε*) = 305 (+2.0), 271 (−9.2), 244 (−16.3), 212 (38.3) nm; for ^1^H NMR, ^13^C NMR, see Table 1; HR-ESI-MS *m/z* 561.2590 [M + H]^+^ (calcd for C_32_H_36_N_2_O_7_, 561.2523 [M + H]^+^).

### 3.4. Quantum-Chemical Calculation

Monte Carlo conformational searches were run with the Spartan 10 software using the Merck Molecular Force Field (MMFF). The Selected conformers, which account for more than 1% of the Boltzmann distribution, were initially optimized at the B3LYP/6-31+G (d,p) level with the CPCM polarizable conductor calculation model in MeOH. The conformers of 1 and 2 were calculated via ECD using the Time-dependent Density functional theory (TD-DFT) method at the B3LYP/6-31+G (d,p) level in MeOH, and the rotational strengths of 30 excited states were calculated. ECD spectra were generated using the SpecDis 1.6 (University of Würzburg, Würzburg, Germany) and GraphPad Prism 5 (University of California San Diego, USA) software by applying Gaussian band shapes with sigma = 0.3 eV from dipole-length rotational strengths [28].

### 3.5. Bioactivity Assays

#### 3.5.1. Cell Culture

HepG2 (ATCC HB-8065) human hepatocellular carcinoma cells were cultured in a MEM (Minimum Essential Medium) growth medium with 10% FBS (foetal bovine serum, TBD Science, Tianjin, China). Cells were cultured at 37 °C with 5% CO_2_.

#### 3.5.2. Cytotoxicity and Proliferation Assay

Cell proliferation was determined by the Cell Counting Kit-8 (CCK-8) method. Cells were seeded in 96-well plates at a density of 2 × 10^5^ cells/mL and cultured for 20 h. Then, cells were treated with various concentrations of compounds or 1% DMSO (vehicle) in MEM supplemented with 10% FBS and cultured for 72 h. *cis*-platin (DDP) (Sigma, St. Louis, MO, USA) was used for the positive control. Then, 10 μL CCK-8 solution was added to each well and incubated for another 1 h at 37 °C to convert WST-8 into formazan. Absorbance was monitored at 450 nm using a microplate reader. All experiments were repeated in triplicate, and the IC_50_ (concentration required to inhibit cell growth by 50%) was determined using the Graphpad Prism 5 software (University of California San Diego, USA).

## 4. Conclusions

Two novel cytochalasan alkaloids, termed Chaetomadrasins A and B (**1**–**2**), together with six known analogues (**3**–**8**), were isolated from the soil-derived fungus, *Chaetomium madrasense* 375. Chaetomadrasins A (**1**) belongs to the chaetoglobosin family and characterized by the presence of all oxygen atoms in the form of Carbonyl for the first time. Chaetomadrasin B (**2**) represents the first example of chaetoglobosin type cytochalasan characterized by a hydroxy group and carbonyl group fused to the indole ring. The absolute configurations of compounds **1**–**2** were determined by extensive analysis of spectroscopic data and quantum chemical ECD calculations. Unfortunately, the antiproliferative activity of compounds **1** and **2** were evaluated against only HepG2 cell lines on base of small amount of the samples and showed moderate antiproliferative activity. In addition, further biological assays and structural diversity of cytochalasans are worth unveiling in future research.

## Figures and Tables

**Figure 1 molecules-24-03240-f001:**
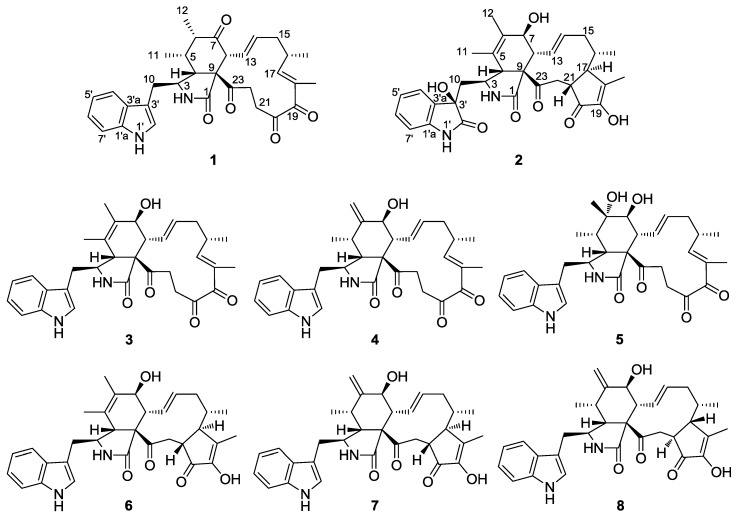
Structures of isolated Compounds **1**–**8**.

**Figure 2 molecules-24-03240-f002:**
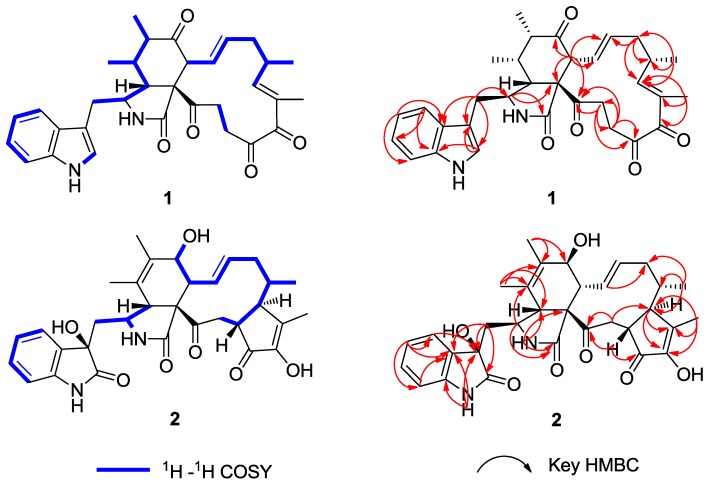
^1^H-^1^H COSY, Key HMBC of Compounds **1**–**2**.

**Figure 3 molecules-24-03240-f003:**
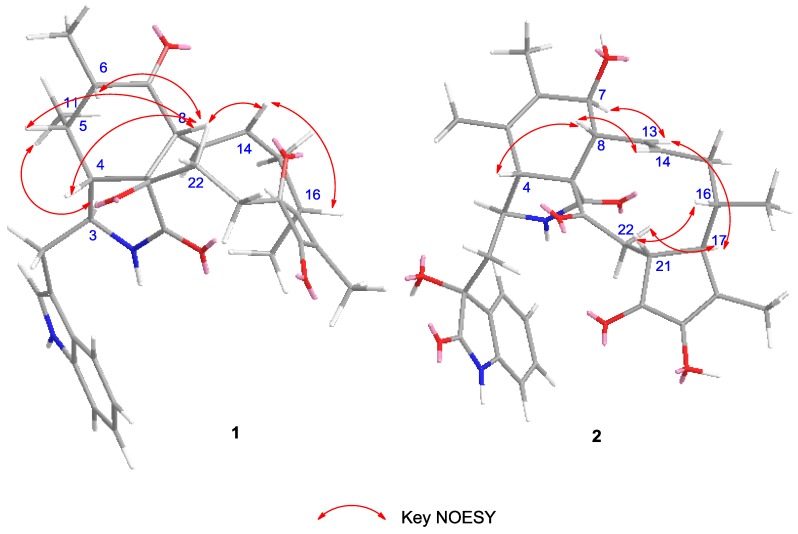
Key NOESY correlations of Compounds **1**–**2**.

**Figure 4 molecules-24-03240-f004:**
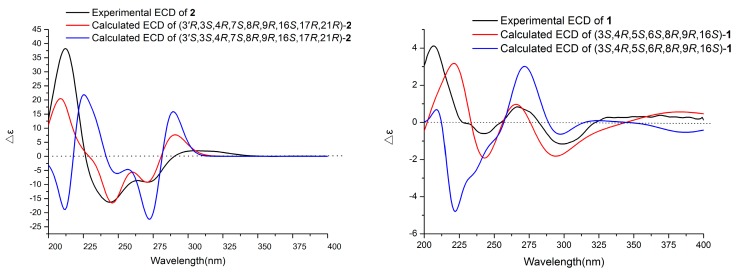
Experimental and calculated ECD spectra of Compounds **1** and **2**.

**Table 1 molecules-24-03240-t001:** ^13^C (100 MHz) and ^1^H (400 MHz) NMR spectroscopic data for Compounds **1** and **2**.

NO.	1 ^a^		2 ^b^	
	*δ*_H_, mult. (*J* in Hz)	*δ*_C_, mult.	*δ*_H_, mult. (*J* in Hz)	*δ*_C_, mult.
1′	8.71, s		10.3, s	
1′a		136.1, C		142.3, C
2′	7.09, s	124.6, CH	6.96, s	178.8, CH
3′		109.2, C		74.6
3′a		127.5, C		131.6
4′	7.51, d (7.0)	118.2, CH	7.22, d (7.2)	124.7, CH
5′	7.15, t (7.4)	120.2, CH	7.00, t (7.4)	122.1, CH
6′	7.22, t (7.5)	122.6, CH	7.23, t (7.0)	129.8, CH
7′	7.36, d (8.0)	111.6, CH	6.82, d (8.0)	110.6, CH
1		174.1, C		174.5, C
2	7.52, s		7.74, s	
3	3.86, m	52.6, CH	3.16, m	52.3, CH
4	2.42, m	46.9, CH	2.80, s	51.6, CH
5	2.23, m	35.4, CH		125.9, C
6	2.14, m	46.1, CH		134.5, C
7		213.4, C	3.79, d (8.7)	68.9, CH
8	3.84, d (9.4)	53.0, CH	2.07, m	52.9, CH
9		62.9, C		65.5, C
10	a: 3.15, m; b: 2.83, m	32.6, CH_2_	a: 1.80, m; b: 1.73, m	43.8, CH_2_
11	1.20, d (6.4)	15.9, CH_3_	1.32, s	16.9, CH_3_
12	1.16, d (6.8)	15.9, CH_3_	1.54, s	15.0, CH_3_
13	5.85, dd (15.1, 9.5)	122.9, CH	6.03, (15.4, 9.8)	131.4, CH
14	5.00, m	135.5, CH	5.05, m	133.0, CH
15	a: 2.35, m; b: 1.90, m	39.7, CH_2_	a: 2.22, m; b: 1.87, m	44.3, CH_2_
16	2.68, m	33.3, CH	1.55, m	42.6, CH
17	6.06, d (9.9)	155.7, CH	2.09, d (5.8)	54.0, CH
18		131.6, C		147.7, C
19		195.7, C		150.0, C
20		204.8, C		203.5, C
21	a: 2.75, m; b: 1.94, m	32.4, CH_2_	2.23, m	50.6, CH
22	a: 2.86, m; b: 1.63, m	36.7, CH_2_	a: 3.20, m; b: 2.00, m	44.0, CH_2_
23		205.6, C		211.6, C
16-Me	1.00, d (6.6)	19.4, CH_3_	0.97, d (6.8)	21.8, CH_3_
18-Me	1.86, s	10.6, CH_3_	1.96, s	17.1, CH_3_
7-OH			4.62, s	
19-OH			9.00, s	
3′-OH			6.06, s	

^a^ Measured in CDCl_3_. ^b^ Measured in DMSO-*d*_6_.

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
