# Peer review of "Chaetomadrasins A and B, Two New Cytotoxic Cytochalasans from Desert Soil-Derived Fungus Chaetomium madrasense 375"

_molecules, 2019, doi:10.3390/molecules24183240_

Round 1
Reviewer 1 Report
The paper describes two new cytochalasins which vary slightly from previously reported compounds. They exhibit modest activity against the HepG2 liver cancer cell line. Compound 2 has an unusual oxidation of the indole ring which makes it different (there are epoxide and open ring analogs of the indole known)
Some small suggestions to improve the paper are:
In the discussion of the structure elucidation for 2, for ease of reading, they should add on page 4 line 91 that chaetoglobosin Vb is 6 and cytoglobosin A is 7. This will help readers to follow the discussion.
For the activity against HepG2 how many replicate experiments (n=3?) were run and what is the standard error? Also do they actually get two decimal place accuracy in their assay protocol? (e.g. 19.4 µM versus 19.40 µM and 8.7 µM rather than 8.67 µM?). They should also list the program they used to calculate the IC50s.
One also wonders if the HPLC separations were accurate to hundredths of minutes. For example armochaetoglobin U is reported to elute with a retention time of 27.02 minutes and others are 30.70 minutes, etc. It is not clear that this level of reproducibility is achievable (there was likely more than one injection done to collect multiple mg on a 1 cm diameter column) and it is difficult to believe that each time they came off at 27.02 minutes. They should round the tR to the nearest tenth of a minute.
What was the source of the HepG2 cells? Typically a catalog number or other source is added for reference and to allow reproduction of the work by other labs.
Page 3 line 72 small grammatical issue - remove the word “found” e.g. in all the known chaetoglobosins, only chaetoglobosin Y and ....
For the HRMS- the data suggest a molecular formula rather than indicate it- typically there is more than one formula consistent with the HRMS data and therefore NMR and other data are coupled with the HRMS to suggest a molecular formula. So on page 2 line 53 change indicate to suggest and on page 4 line 85- similar comment change to say a Molecular formula of X was suggested by the HR-ESI-MS. There is clearly more than one possible formula for this observed accurate mass.
Author Response
Referee 1
In the discussion of the structure elucidation for 2, for ease of reading, they should add on page 4 line 91 that chaetoglobosin Vb is 6 and cytoglobosin A is 7. This will help readers to follow the discussion.
Response: the numbers of chaetoglobosin Vb and cytoglobosin A have been added.
For the activity against HepG2 how many replicate experiments (n=3?) were run and what is the standard error? Also do they actually get two decimal place accuracy in their assay protocol? (e.g. 19.4 µM versus 19.40 µM and 8.7 µM rather than 8.67 µM?). They should also list the program they used to calculate the IC50s.
Response: All experiments were repeated in triplicate and the IC50 (concentration required to inhibit cell growth by 50%) was determined using Graphpad Prism 5 software. We supplied the wrong decimal place, it should be 19.4 µM and 8.7 µM.
One also wonders if the HPLC separations were accurate to hundredths of minutes. For example armochaetoglobin U is reported to elute with a retention time of 27.02 minutes and others are 30.70 minutes, etc. It is not clear that this level of reproducibility is achievable (there was likely more than one injection done to collect multiple mg on a 1 cm diameter column) and it is difficult to believe that each time they came off at 27.02 minutes. They should round the tR to the nearest tenth of a minute.
Response: the tR have been corrected to the nearest tenth of a minute.
What was the source of the HepG2 cells? Typically a catalog number or other source is added for reference and to allow reproduction of the work by other labs.
Response: the deposit number of HepG2 cells has been supplied.
Page 3 line 72 small grammatical issue - remove the word “found” e.g. in all the known chaetoglobosins, only chaetoglobosin Y and.
Response: the word “found” has been deleted.
For the HRMS- the data suggest a molecular formula rather than indicate it- typically there is more than one formula consistent with the HRMS data and therefore NMR and other data are coupled with the HRMS to suggest a molecular formula. So on page 2 line 53 change indicate to suggest and on page 4 line 85- similar comment change to say a Molecular formula of X was suggested by the HR-ESI-MS. There is clearly more than one possible formula for this observed accurate mass.
Response: the word “indicate” has been replaced by suggest.
Reviewer 2 Report
The authors of Chen et al, describe the isolation and characterization of a fungus Chaetomium madrasense strain from the soil collected from Hotan City in China. The genus and species were identified phenotypically and genotypically. The strain was fermented rice media for four weeks and the organic extract was prepared and subjected to fractionation to provide a group of cytochalasans. Two of the cytochalasans, chaetomadrasins A and B are new compounds and the structures were determined 1- and 2D NMR spectroscopy. The relative configuration was deduced using NOESY and the absolute configuration was determined using ECD.
The article is very interesting in topics related to natural products chemistry. It is well written and thus I recommend this article to be published with the following major edits/comments:
The authors should compare the HPLC chromatogram of the crude extract with that of the pure compounds 1 and 2 to make sure that 2 is not the reduction product of 1 during work out.
Please include 1H and 13C NMR spectra of compounds 3-8 in the supplementary information
For the biological assay, all compounds 1-8 should be tested against HepG2 instead of just 1 and 2. The data will provide information on the structure-activity relationship and the authors should comment on this.
Line 29: should read “These fungal alkaloids are characterized by …..”
Author Response
Referee 2
The authors should compare the HPLC chromatogram of the crude extract with that of the pure compounds 1 and 2 to make sure that 2 is not the reduction product of 1 during work out.
Response: The HPLC chromatogram of the crude extract with that of the pure compounds 1 and 2 have been supplied (see Fig. S26).
Please include 1H and 13C NMR spectra of compounds 3-8 in the supplementary information.
Response: 1H and 13C NMR spectra of compounds 3-8 have been supplied (see Fig. S27-40).
For the biological assay, all compounds 1-8 should be tested against HepG2 instead of just 1 and 2. The data will provide information on the structure-activity relationship and the authors should comment on this.
Response: the cytotoxic activities (against H292, HCT116, P388, A549, KB, HL-60, SMMC-7721, MCF-7, and SW-480 cell lines) of 3-8 have been previously reported, and all have no remarkable activity. so their antibacterial activities were tested, unfortunately compounds 3-8 were inactive against Escherichia coli CGMCC 1.2836, Shewanella putrefaciens CGMCC 1.3667 and Pseudomonas aeruginosa CGMCC 1.860 (MIC ≥ 100 μg/ml), and all the compounds have no sufficient weight for further cytotoxicity testing.
Line 29: should read “These fungal alkaloids are characterized by …..”
Response: the word “contributed” has been replaced by “characterized”.
Reviewer 3 Report
In this work the authors reported the isolation of eight alkaloids among which two new molecules were fully characherized from a chemical point of view. Moreover, these new alkaloids were biologically evaluated for their antiproliferative activity against HepG2 cell line.
Although the manuscript is clearly written and well organized, it seems weakly developed from a biological point of view. As mentioned above, the two new compounds were tested against HepG2 cells whose choice is not supported by scientific literature in the text. Moreover, in order to explore the cytotoxic activity of the new compounds and in order to make the manuscript suitable for publication, further experiments are needed in additional cell lines and to identify a possible mechanism of action.
Author Response
Referee 3
Although the manuscript is clearly written and well organized, it seems weakly developed from a biological point of view. As mentioned above, the two new compounds were tested against HepG2 cells whose choice is not supported by scientific literature in the text. Moreover, in order to explore the cytotoxic activity of the new compounds and in order to make the manuscript suitable for publication, further experiments are needed in additional cell lines and to identify a possible mechanism of action.
Response: the analogues of the two new compounds were tested against tumor cells whose choice is supported by scientific literature (see Introduction section) in the text. So we selected one of tumor cells HepG2 as the testing object. At the same time, the antibacterial activities of the two new compounds also were tested, unfortunately compounds 1 and 2 were inactive (MIC ≥ 100 μg/ml) against Escherichia coli CGMCC 1.2836, Shewanella putrefaciens CGMCC 1.3667 and Pseudomonas aeruginosa CGMCC 1.860, and the compounds have no sufficient weight for further possible mechanism of action.
Round 2
Reviewer 2 Report
The authors made all the required changes. I recommend the paper for publication in the current form.
Author Response
We wish to express our appreciation for your in-depth comments, suggestions, and corrections, which have greatly improved the manuscript.
Reviewer 3 Report
The perplexities relataed to the previous version of the manuscript remain. The authors did not explain why tested the compounds just only on that cell line and did not reported data on the antiproliferative activity on other cancer cell lines. Thus, I cannot recommend the publication of this manuscript on Molecules.
Author Response
Response: on base of the small amount of the samples, it is not enough to evaluate the antiproliferative activity on other cancer cell lines, and made the fermentation and isolation(repeat it) again is not on the table, the suggestion of the reviewers’ comments is necessary and meaningful, further biological assays are worth unveiling in future research.